# Thermal Lensing and Laser-Induced Damage in Special Pure Chalcogenide $Ge_{35}As_{10}S_{55}$ and $Ge_{20}As_{22}Se_{58}$ Glasses under Quasi-CW Fiber Laser Irradiation at 1908 nm

Oleg Antipov [1,*], Anton Dobrynin [1,2], Yuri Getmanovskiy [1,3], Ella Karaksina [4], Vladimir Shiryaev [4], Maksim Sukhanov [4] and Tatiana Kotereva [4]

1. Institute of Applied Physics of the Russian Academy of Sciences, 603950 Nizhny Novgorod, Russia
2. Radiophysics Department, Nizhny Novgorod State University, 603022 Nizhny Novgorod, Russia
3. Department of Material Sciences and Technologies, Nizhny Novgorod State Technical University, 603950 Nizhny Novgorod, Russia
4. Institute of Chemistry of High-Purity Substances of the Russian Academy of Sciences, 603951 Nizhny Novgorod, Russia
* Correspondence: antipov@ipfran.ru; Tel.: +7-9038463849

**Abstract:** Special pure chalcogenide glass is the material of choice for many mid-infrared optical fibers and fiber lasers. In this paper, the thermo-optical lensing and laser-induced damage were studied in $Ge_{35}As_{10}S_{55}$ and $Ge_{20}As_{22}Se_{58}$ glasses and compared with the well-studied $As_2S_3$ glass. The thermal Z-scan technique with the quasi-CW Tm-doped fiber laser at 1908 nm was applied to study thermal lensing in chalcogenide glass. The laser-induced damage of various chalcogenide glasses was determined using the one-on-one procedure. The thermal nonlinear refractive index of the $Ge_{35}As_{10}S_{55}$ and $Ge_{20}As_{22}Se_{58}$ glasses was found to be lower than that of the $As_2S_3$ glass. The laser-induced damage threshold of the $Ge_{20}As_{22}Se_{58}$ glass was determined to be higher than that of the $Ge_{35}As_{10}S_{55}$ glass. The difference in the thermal damage threshold of the $Ge_{35}As_{10}S_{55}$ and $Ge_{20}As_{22}Se_{58}$ glasses and their lower value in comparison with the $As_2S_3$ glass were explained by a deviation from the stoichiometry of glass compositions and their tendency to crystallize.

**Keywords:** chalcogenide glass; thermo-optical effects; thermal lenses; Z-scan testing; laser-induced damage threshold

## 1. Introduction

Chalcogenide glass (ChG) is a promising material for the development of mid-infrared (mid-IR) optical fibers, fiber lasers and other photonic devices. ChGs have unique properties, such as high transmission in a wide range of wavelengths (from visible to far IR), low phonon energy, relatively high chemical resistance to environmental factors, the ability to dope with rare earth metal ions, as well as high nonlinearity, good photosensitivity, and a number of other characteristics. Achieving the high-brightness mid-IR broadband supercontinuum (SC) generated in nonlinear chalcogenide waveguide was notable [1–3]. Recently, lasing in the 4.5–5.9 μm wavelength range was demonstrated in glass samples and optical fibers based on ChGs doped with $Ce^{3+}$, $Tb^{3+}$, and $Pr^{3+}$ ions [4,5]. The CW Tm-fiber laser pumping at ~1.9 μm was used, for example, for the mid-IR fiber lasers based on $Tb^{3+}$-doped ChGs [4–6]. The power scaling of the ChGs lasers is limited by the laser-induced damage [5,7]. A number of researchers have studied the failure mechanisms in such materials and evaluated the laser-induced damage threshold (LIDT) in various conditions [8–10]. Similar studies are also relevant for other problems related to the development of optical materials based on ChGs and capable of transmitting high-power IR radiation.

The resistance of laser materials to the laser-induced damage is determined by several factors. First is chemical composition. In recent years, multiple studies have been done to

investigate the three-component ChGs (Ge-As-S, Ge-As-Se systems) and the effect of glass composition at various irradiation settings [4,5]. It has been found that the germanium doping increases by over 2× LIDT induced by the femtosecond laser pulses compared to the commercially available and previously studied $As_2S_3$ and $As_2Se_3$ [8–10]. Higher LIDT in Ge-doped chalcogenide glass is attributed to the fundamental glass characteristics and, in particular, to the strong bonds of Ge atoms in the material. However, the mechanism of the laser-induced damage notably depends not only on the material parameters but also on the irradiation wavelength and the lasing regimes, such as pulse width, repetition rate, peak intensity or energy fluence [11,12]. Unlike optical damage under near-IR femtosecond pulses, the damage mechanism under the mid-IR long pulses (with the pulse width of a hundred nanoseconds or more) is mainly attributed to the temperature buildup rather than the accumulation of conduction band electrons [11,12].

The presence of optical inhomogeneities in the ChGs that can act as IR absorption or scattering centers can also strongly decrease the LIDT [11,13]. Such inhomogeneities can include, for example, waviness, bubbles, impurities, inclusions of crystalline phases, and nonstoichiometry. The concentration of these inhomogeneities, as a rule, depends on the method of glass preparation, which includes the stages of synthesis and purification of each glass component and the final glass. It appears that insufficient attention has been paid to this problem in the literature to date. At the same time, the influence of the purity and optical homogeneity of glass on the laser damage threshold is undoubtably important for the study of candidate materials for the transmission of high-power IR radiation.

Strong laser-induced thermo-optical effects in ChGs (optical distortions by heated windows, thermal lenses and deformations) were reported in [14–16]. These effects decreased the LIDT in the ChG fibers and waveguides under the laser irradiation with different pulse width. However, many thermo-optical parameters of the some ChGs are still undetermined [10,15–17].

This paper presents the results of studies of the thermo-optical lensing and laser-induced damage of special pure $Ge_{35}As_{10}S_{55}$ and $Ge_{20}As_{22}Se_{58}$ glasses under the quasi-CW irradiation with the Tm-doped fiber laser at 1908 nm. The results are then compared to the $As_2S_3$ glass. The ChGs samples were obtained using the techniques developed by the authors of this article.

## 2. Glass Preparation

The $Ge_{35}As_{10}S_{55}$ and $Ge_{20}As_{22}Se_{58}$ glasses with a high content of germanium were chosen for the laser damage resistivity studies due to their sufficiently high glass transition temperature and improved thermal characteristics. We used the $Ge_{35}As_{10}S_{55}$ and $Ge_{20}As_{22}Se_{58}$ glass compositions as core or cladding materials for the fabrication of step-index optical fibers doped by rare earth or Bi ions due to their suitable optical and thermal properties, glass transition temperature, refractive index, and low adhesion to silica glassware. Previously [18], step-index optical Bi-doped fiber based on $Ge_{35}As_{10}S_{55}$ glass was fabricated; the ability to draw fiber indicates the low tendency to crystallization.

For the glass synthesis, germanium, arsenic, sulfur, and selenium with a purity of 6 N were used. Chalcogens were additionally purified from hydrogen, carbon, and oxygen impurities. The content of these impurities, as a rule, is not controlled in commercial samples. Sulfur was subjected to vacuum distillation combined with the chemical-thermal treatment using cerium(IV) oxide at 800–900 °C [19]. Selenium was melted with selenium(IV) oxide and subjected to double vacuum distillation at a low evaporation rate to remove oxygen, silicon, carbon, and heterophase inclusions [20,21]. The germanium granules were calcined in vacuum at 700 °C for 2 h to remove the GeO surface film. Arsenic was purified by vacuum sublimation.

Special pure $Ge_{35}As_{10}S_{55}$ and $Ge_{20}As_{22}Se_{58}$ glasses were produced in silica-glass reactors using chemical distillation purification of glass-forming melts with aluminum(III) chloride to remove oxygen and hydrogen impurities [4]. Next, the glass-forming melt was purified by vacuum distillation (2 times in dynamic vacuum and 1 time in a closed system).

The final stage, the glass homogenization, was carried out in a muffle rocking furnace for 7 h at 850 °C and 800 °C for $Ge_{35}As_{10}S_{55}$ and $Ge_{20}As_{22}Se_{58}$, respectively. The melt was solidified by air-quenching, with subsequent glass annealing and slow cooling to room temperature. $As_2S_3$ glass was prepared by direct melting of the arsenic monosulfide and elementary sulfur in a sealed silica ampoule.

The transmission spectra of the studied ChGs with a 10 mm optical path length were recorded using the Fourier-transform IR spectrometer IRP Prestige–21 (Shimadzu, Japan) in the spectral range 7000–350 cm$^{-1}$. Figure 1 shows the absorption coefficients of $Ge_{20}As_{22}Se_{58}$ (1) and $Ge_{35}As_{10}S_{55}$ (2) glasses in the 1.5–8 μm wavelength range. For clarity, the spectra are separated by different heights along the ordinate axis, that is, the value of the absorption coefficient (cm$^{-1}$) plus a constant (cm$^{-1}$) is plotted along the *y*-axis. The spectra demonstrate the transmission region of the samples, the presence of optically active impurities, and the magnitude of the absorption coefficient due to each impurity. The volume absorption coefficient of glasses, measured at 1.975 μm by laser calorimetry, was about 10$^{-3}$ cm$^{-1}$.

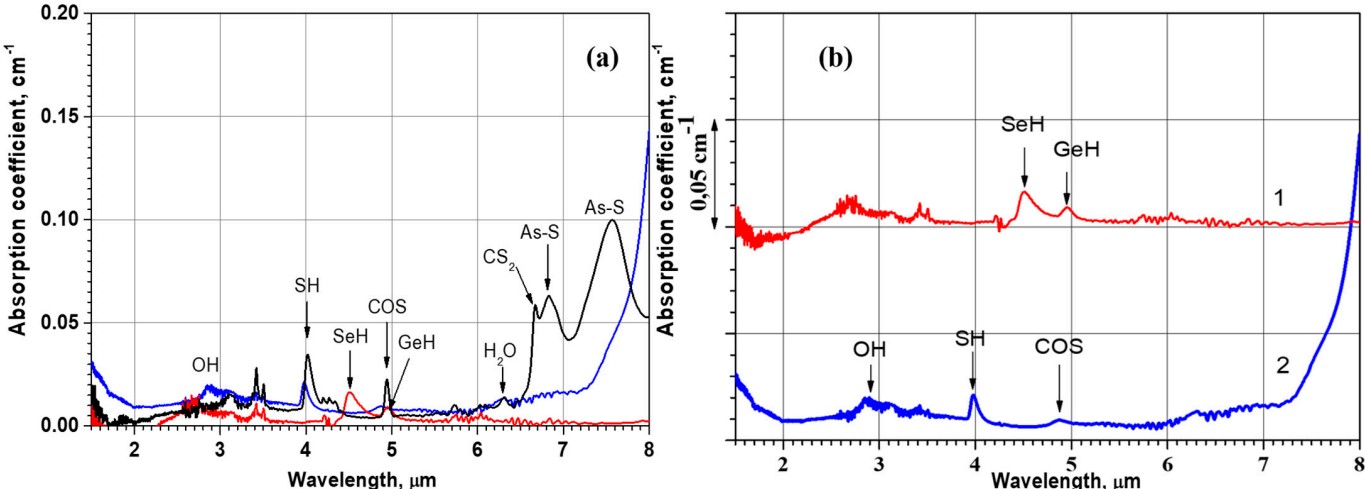

**Figure 1.** Mid-IR absorption spectra of ChGs: $As_2S_3$ (black), $Ge_{35}As_{10}S_{55}$ (blue), $Ge_{20.5}As_{22}Se_{58}$ (red) on the same figure (**a**) or separated (**b**).

It can be seen that the spectrum 1 of the $Ge_{20}As_{22}Se_{58}$ glass (Figure 1b) did not exhibit strong absorption bands corresponding to the dissolved limiting gas-forming impurities, with the exception of low-intensity SeH and GeH bands at 4.57 and 4.9 μm, respectively. The spectrum 2 for $Ge_{35}As_{10}S_{55}$ glass contains low-intensity absorption bands of OH, SH, and COS impurities. There are no intense impurity As-O and Ge-O bands in the absorption spectra of the glasses. The spectrum of the reference $As_2S_3$ glass also doesn't contain strong impurity absorption bands.

From the mid-IR absorption coefficient spectra, the content of gas-forming impurities (hydrogen in the form of SeH- and SH- bonds, oxygen in the form of As-O and Ge-O bonds) was determined using known extinction coefficients [22–24]. In the $Ge_{20}As_{22}Se_{58}$ glass, the estimated hydrogen content, in the form of a SeH-group, was 0.06 ppmw; the estimated content of oxygen in the Ge-O and As-O forms was 0.16 and 3.5 ppmw, respectively. In the $Ge_{35}As_{10}S_{55}$ glass, the estimated hydrogen content, in the form of SH groups, was 0.04 ppmw; the content of COS impurity was 8 ppb(mol). These concentrations of gas-forming impurities were not a critical factor for achieving low radiation absorption in the IR. The content of micro- and sub-micro-inclusions in the samples, as well as waviness, was controlled by laser ultra-microscopy with an infrared camera to exclude or minimize their influence on the results of the LIDT experiments.

The thermal properties of ChGs were analyzed using differential scanning calorimetry (DSC). DSC was carried out in the 100–550 °C temperature range using a synchronous Netzsch STA 409 PC Luxx analyzer with the sensitivity of 1 μV/mW and ±0.5 K temper-

ature accuracy, at a heating rate of 10 K/min. The glass transition temperature ($T_g$) was determined from the intersection points of tangents in the initial region of thermal events. DSC curves are shown in Figure 2. Curves illustrate only one peak corresponding to the glass transition temperature; there is no peak associated with crystallization for both types of glass.

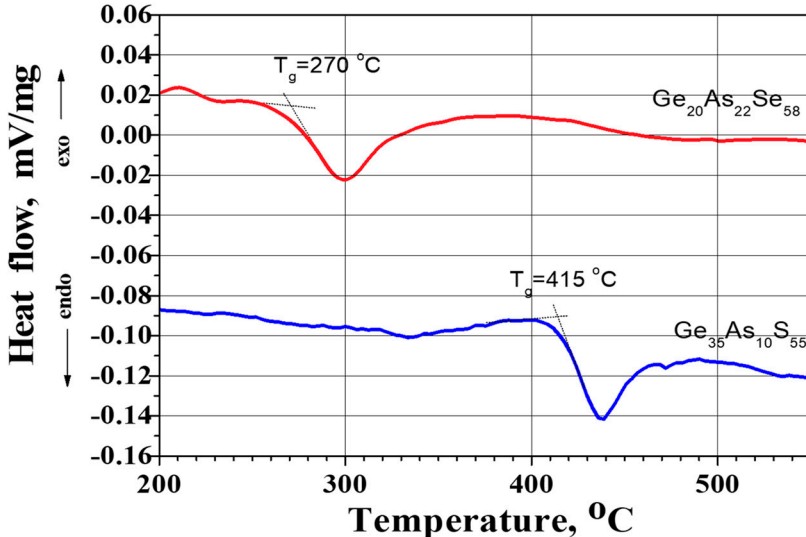

**Figure 2.** DSC thermograms for $Ge_{20}As_{22}Se_{58}$ (red) and $Ge_{35}As_{10}S_{55}$ (blue) glasses at a heating rate of 10 K/min.

Glass samples were made in the form of 12–15 mm diameter disks with 2–2.5 mm thickness. The working surfaces were slanted at ~5° to divert the reflections and to decrease the impact of the laser beam interference on the laser-induced damage of the input surface. The quality of the polished surfaces of the samples was controlled by optical microscopy.

### 3. Experimental Techniques and Results

#### 3.1. Thermal Lens Z-Scanning

The nonlinear thermo-optical properties of the material, specifically, the thermal lenses induced in the ChG by the Tm fiber laser, were tested using the Z-scan technique [25–31]. This technique is based on the measurements of the transmitted signal of a ChG sample that is moved along the longitudinal *z*-axis of a focused laser beam. Changes in the refractive index due to optical nonlinearities are translated into changes in the laser beam dimension at the detector plane. These changes can be measured using a detector placed behind a small aperture. The transmitted signal depends on the sample position, and by measuring variations in the transmitted signal one can estimate the nonlinear refractive index.

The Tm fiber laser "LTM–50" ("NTO IRE-Polus", Fryazino, Moscow region, Russia) provided the single-longitudinal-mode beam ($M^2 < 1.1$) at 1908 nm with CW power up to 55 W. The pump current of the CW fiber laser was modulated to provide controllable pulses with a pulse width of 5–10 s. The fiber laser beam was focused by a lens with the focal length of 10.5 cm. The Rayleigh distance of the focused beam was $z_R \approx 13$ mm. The ChG disc was placed near the focal point, and its position was scanned along the axis of the focused laser beam (Figure 3). A pinhole with an aperture diameter of 0.5 mm was placed in the far-field with respect to the disc position (the distance between the disc and pinhole was ~70 cm; the initial transmission of the aperture was ~10% of the incident power). The power of the transmitted beam was measured by a power meter, PM (Ophir "30(150)A-BB-18ROHS" power sensor with "StarBright" power meter). The Ophir-Spiricon "PYROCAM IV" beam profiler was also used to analyze the full-beam structure before the small aperture.

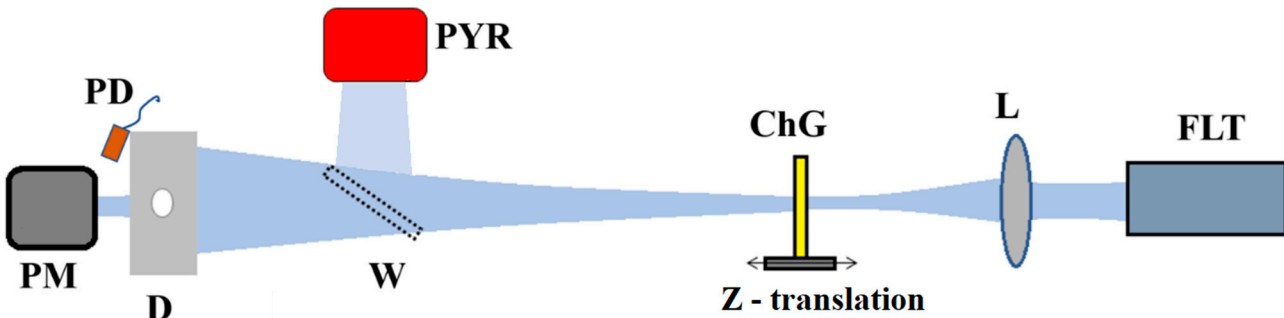

**Figure 3.** Schematic of the thermal lens z-scanning. FLT is the fiber laser telescope, L is the focusing lens, ChG is the disc on the z-translation stage, W is the wedged plate, PYR is the PYROCAM IV camera, D is the pinhole with a small aperture, PM is the power meter, PD is the photodiode.

The transmitted laser beam was tested to avoid strong thermal lens aberrations. Initially, the laser power was determined, which didn't lead to the appearance of aberration rings (Figure 4a). Then, the transient signal kinetic was detected by HAMAMATSU InGaAs PIN G8422-03 photodiode, PD (a beam chopper between the fiber laser telescope and the lens was used for the kinetic study).

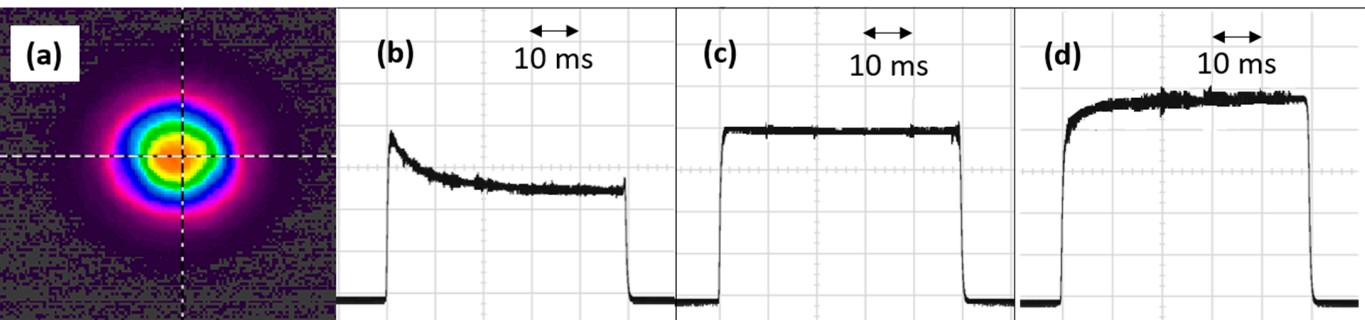

**Figure 4.** The PYROCAM image of the laser beam transmitted through the ChG sample (**a**). The PD-registered waveforms for different position of the $As_2S_3$ disk with respect to the focal plane: before the focal point (**b**), in the focal point (**c**), and after the focal point (**d**).

The registered kinetics demonstrated the thermal self-focusing behavior of the laser beam transmitted through the ChG discs: the on-axis intensity had the characteristic "relaxation" time of about 20–30 ms that was determined by the thermal lens kinetics (Figure 4b,d). For the long pulses with pulse width $\tau_p$, which was much longer than the time $\tau_D$ of the thermal diffusion on the focused-beam diameter, the thermal lens reached the steady-state condition. The exposure duration (the pulse width) for the Z-scan was set at $\tau_p$ = 5–10 s, which was much longer than the characteristic thermal diffusion time on the beam waist in ChG, $\tau_D \approx 4w_0^2 \rho C_p / K_T \approx 10$ ms (where $w_0$ is the focal beam radius; $\rho$, $C_p$, and $K_T$ are the density, the specific heat capacity, and the thermal conductivity of the ChG, respectively). Due to this difference, we can say that the on-axis normalized transmittance $T(Z)$ was determined as a function of the sample position $z$ in a steady-state regime. The measured power was normalized to the power measured in the absence of the thermal lens. The experimentally measured steady-state transmittance $T(Z)$ of three ChGs samples for different fiber laser power is shown in Figure 5.

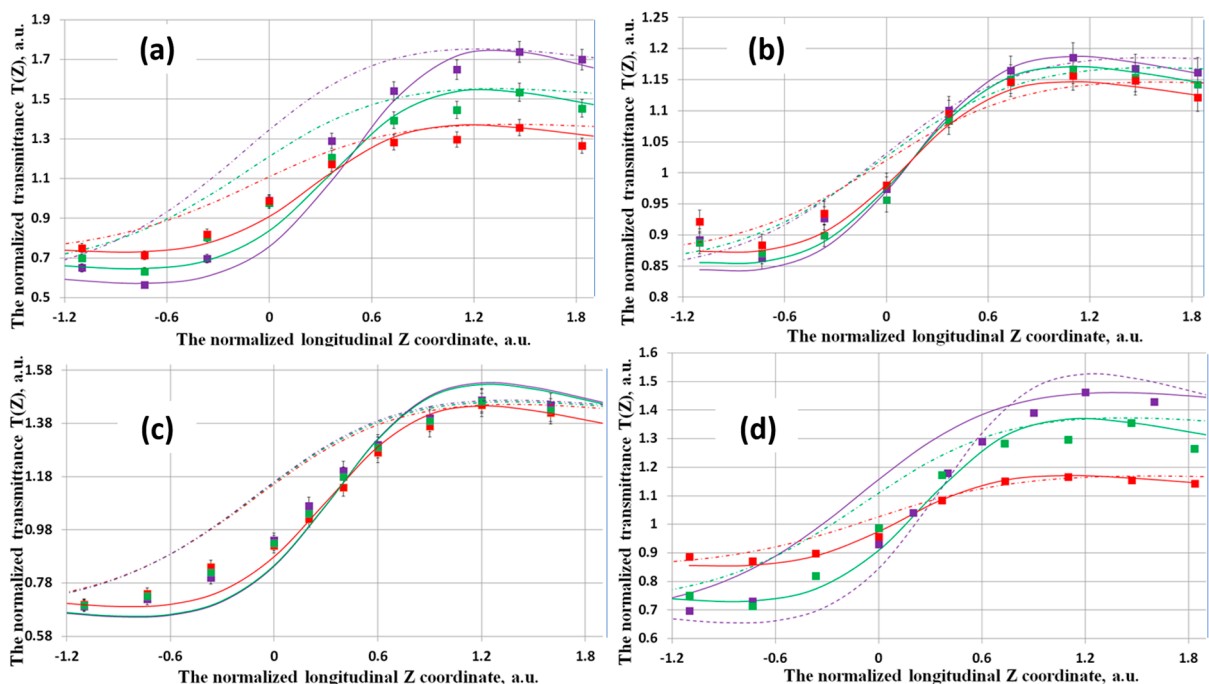

**Figure 5.** The normalized transmittance vs. normalized position of the ChG discs in the beam waist, Z. (**a**) $Ge_{35}As_{10}S_{55}$ irradiated with 1.31 W (red), 2.055 W (green), 2.77 W (violet) of the incident power; (**b**) $Ge_{20.5}As_{22}Se_{58}$ irradiated with 0.79 W (red), 1.34 W (green), 1.71 W (violet) of the incident power; (**c**) $As_2S_3$ element irradiated with 0.69 W (red), 1.07 W (green), 1.47 W (violet) of the incident power; (**d**) comparative curves for the $Ge_{35}As_{10}S_{55}$ disc at 1.31 W power (red), the $Ge_{20.5}As_{22}Se_{58}$ disc at 1.71 W power (green), and the $As_2S_3$ disc at 1.47 W power (violet). The solid-fitting curves are the parabolic thermal lens approximations, the dashed curves are the aberrant thermal lenses.

The theoretical $T(Z)$ curves in Figure 5 were modeled using two different expressions in accordance with two thermal lens models:

$$T(Z) = \left(1 - 2\theta Z / \left(1 + Z^2\right) + \theta / \left(1 + Z^2\right)\right)^{-1} \tag{1}$$

for the parabolic thermal lens approximation [26,28], and

$$T(Z) = 1 + \theta arctg\left(\frac{2Z}{3 + Z^2}\right) + \left(\frac{\theta}{2}arctg\left(\frac{2Z}{3 + Z^2}\right)\right)^2 + \left(\frac{\theta}{4}\ln\left(\frac{1 + Z^2}{9 + Z^2}\right)\right)^2 \tag{2}$$

for the aberrant thermal lens [30]; where $Z = z/z_R$ assuming the distance $L_a$ between the sample and aperture is much larger than the Rayleigh range, $L_a \gg z_R$, the point $z = 0$ is the focus point, $\theta$ is the on-axis phase shift given by expression:

$$\theta = \frac{\alpha L P \xi}{\lambda K_T}\left(\frac{\partial s}{\partial T}\right) \tag{3}$$

where $\alpha$ is the absorption coefficient, $P$ is the power of the laser beam transmitted into the ChG disc, $\lambda$ is the laser wavelength, $\xi$ is the fraction of the absorbed power converted into heat; $(\partial s/\partial T)$ is the temperature coefficient of the optical path length change of the ChG given by expression:

$$\left(\frac{\partial s}{\partial T}\right) \approx \left(\frac{\partial n}{\partial T}\right) + (n_0 - 1)\beta_T(1 + \nu) \tag{4}$$

where $(\partial n/\partial T)$ is the thermo-optical coefficient, $n_0$ is the refractive index, $\beta_T$ is the thermal expansion coefficient, $\nu$ is the Poisson ratio. A similar $T(Z)$ expression as (2), excluding the last term, was also used in several papers [27,29].

Both thermal lens models give the transmittance expressions dependent on the on-axis phase shift θ only. The theoretical curves $T(Z)$ were plotted to provide good fitting of the experimental data (Figure 5).

The main parameter determined by the comparison of the experimental data with the theoretical model is the on-axis phase shift θ (Table 1). The determined θ value allowed to estimate the steady-state nonlinear thermal refractive index $n_{2T}$ can be expressed as:

$$n_{2T} = \frac{\lambda \theta w_0^2}{4PL} \tag{5}$$

**Table 1.** The determined values of the on-axis phase shift θ and the power-normalized phase shift $\theta/P$ and the nonlinear thermal refractive index $n_{2T}$ for different ChGs at different laser power. Each value of θ and $n_{2T}$ was determined for the parabolic thermal lens (Expression (1))/the aberrant thermal lens (Expression (2)).

| Power, W | Ge$_{35}$As$_{10}$S$_{55}$ | | | Ge$_{20.5}$As$_{22}$Se$_{58}$ | | | As$_2$S$_3$ | | |
|---|---|---|---|---|---|---|---|---|---|
| | θ | $\theta/P$, W$^{-1}$ | $n_{2T}$, cm$^2$/W ($\times 10^{-9}$) | θ | $\theta/P$, W$^{-1}$ | $n_{2T}$, cm$^2$/W ($\times 10^{-9}$) | θ | $\theta/P$, W$^{-1}$ | $n_{2T}$, cm$^2$/W ($\times 10^{-9}$) |
| 0.69 | | | | | | | 0.37/0.7 | 0.54/1.02 | 10.6/20.6 |
| 1.07 | | | | | | | 0.42/0.72 | 0.39/0.67 | 8.0/13.8 |
| 1.47 | | | | | | | 0.43/0.73 | 0.29/0.49 | 6.0/10.1 |
| 0.79 | | | | 0.14/0.26 | 0.18/0.33 | 3.4/6.6 | | | |
| 1.34 | | | | 0.16/0.3 | 0.12/0.22 | 2.6/4.8 | | | |
| 1.71 | | | | 0.17/0.32 | 0.1/0.19 | 2.0/3.8 | | | |
| 1.31 | 0.32/0.6 | 0.25/0.46 | 4.8/9.2 | | | | | | |
| 2.05 | 0.44/0.84 | 0.22/0.41 | 4.2/8.2 | | | | | | |
| 2.77 | 0.57/1.07 | 0.21/0.38 | 4.0/7.8 | | | | | | |

Expression (5) allows the estimation of $n_{2T}$ for the known values of $\lambda$, $w$, $P$, and $L$ (Table 1).

It should be noted that the determined values of the on-axis phase shifts and the nonlinear thermal refractive index $n_{2T}$ depend on the thermal lens model; the aberrant thermal lens model provides almost 2 × the values of the parabolic thermal lens model. This result agrees well with the previous comparison of the thermal lens models [30].

In accordance with Expressions (3)–(5), the nonlinear thermal refractive index $n_{2T}$ depends on the material parameters of the ChG only. However, the experimentally determined $\theta/P$ and $n_{2T}$ values decreased with increase in the laser power. This discrepancy can be explained by a thermal dependence of the thermo-optical coefficient and the thermal conductivity, and also an asigmatic deformation of the absorbed beam transmitted through the wedged ChG discs. Nevertheless, the smallest nonlinear thermal refractive index $n_{2T}$ was determined for the Ge$_{20.5}$As$_{22}$Se$_{58}$ glass ($2.7 \pm 0.7 \times 10^{-9}/5.2 \pm 0.7 \times 10^{-9}$ cm$^2$/W); the medium value $n_{2T}$ had the Ge$_{35}$As$_{10}$S$_{55}$ glass ($4.4 \pm 0.4 \times 10^{-9}/8.5 \pm 0.7 \times 10^{-9}$ cm$^2$/W); the biggest $n_{2T}$ value had the As$_2$S$_3$ glass ($8.3 \pm 1.3 \times 10^{-9}/1.5 \pm 0.5 \times 10^{-8}$ cm$^2$/W).

The $n_{2T}$ coefficient can be theoretically estimated by the following expression:

$$n_{2T} = \frac{\alpha \xi w^2}{4K_T} \left( \frac{\partial s}{\partial T} \right) \tag{6}$$

By using the well-known thermal and thermo-optical parameters of As$_2$S$_3$ glass (Table 2), the $n_{2T}$ coefficient can be calculated as $9.6 \times 10^{-9}$ cm$^2$/W using Expression (5) for a beam radius of 90 μm (at the absorption coefficient of $\alpha\xi = 1.5 \times 10^{-3}$ cm$^{-1}$). This value is in good agreement with the experimentally determined parameter (see Table 1).

**Table 2.** The thermal and thermo-optical parameters and refractive indexes of ChGs.

| | $Ge_{35}As_{10}S_{55}$ | $Ge_{20.5}As_{22}Se_{58}$ | $As_2S_3$ |
|---|---|---|---|
| Thermal conductivity, $K_T$, W/(m·K) | 0.153 [31] | 0.2–0.3 [14] | 0.17 [32] |
| Specific heat capacity, $C_P$, J/(g K) | | 0.33 [14] | 0.46 [32] |
| Density, $\rho$, g/cm³ | 2.96 [31] | 4.4–4.52 [14] | 3.20 [33] |
| Refractive index, $n_0$ | 2.315 at 1.55 μm [17] 2.239 at 6 μm [34] | 2.53 at 4.5 μm [14] 2.631 at 1.8 (for $Ge_{17}As_{20}Se_{58}$) [32] | 2.43 at 1908 nm [33] |
| Thermo-optical coefficient, $(\partial n/\partial T)$, K⁻¹ | | | $7.52 \times 10^{-6}$ K⁻¹ at 1908 nm [35] |
| Thermal expansion coefficient, $\beta_T$, K⁻¹ | | $17.4 \times 10^{-6}$ [14] | $25 \times 10^{-6}$ K⁻¹ [36] |
| Temperature coefficient of the optical path length change, $(\partial s/\partial T)$, K⁻¹ | | $3.8 \times 10^{-5}$ at 4.5 μm [14] | $5.39 \times 10^{-5}$ at 4.5 μm [35,36] |

The lower value of the $n_{2T}$ coefficient of the $Ge_{35}As_{10}S_{55}$ and $Ge_{20.5}As_{22}Se_{58}$ glass in comparison with the $As_2S_3$ glass can be explained by the lower thermo-optical coefficient or the higher thermal conductivity of the $Ge_{20.5}As_{22}Se_{58}$ glass, rather than the lower absorption at 1908 nm.

### 3.2. Laser-Induced Damage Testing

The laser-induced damage of the ChGs was found using the setup described in the previous chapter excluding the small-aperture pinhole (Figure 1). To prevent any impact of the dust particles on the LIDT, both working surfaces of each sample were put into the purified laminar air flow. The one-on-one procedure was used to determine the LIDT [37]. In the one-on-one test, each unexposed site on the surface of the sample was exposed to a single laser pulse with the defined beam parameters (the fixed fluence of the pumping beam at the fixed power and exposure duration). The sample was moved in the surface plane by a positioner over the distance of 2–3 mm, which significantly exceeded the beam diameter. The number of the independent irradiated sites, $N_{IR}$, determined the statistical accuracy of LIDT. The one-on-one procedure provides more accurate LIDT estimations compared to other test procedures ("S-on-1" or "R-on-1") [37–42]. The damage probability was calculated as the ratio of the number of damaged sites $N_D$ to the number of total irradiated sites:

$$Pr = N_D/N_{IR}. \tag{7}$$

The exposure duration (the pulse width) in our experiments was 5 or 10 s. The clear indications of the LIDT breach were the decrease in the transmitted power, the disappearance of a beam image on the PYROCAM and the characteristic noise from the damaged area.

$Ge_{35}As_{10}S_{55}$ glass had the lowest LIDT, while the $Ge_{20.5}As_{22}Se_{58}$ sample had the higher damage threshold (Figure 6). The $As_2S_3$ sample wasn't damaged at the tested laser power of up to 52 W and the beam intensity of up to ~0.37 MW/cm². Both $Ge_{20.5}As_{22}Se_{58}$ and $Ge_{35}As_{10}S_{55}$ samples had the same LIDT when the exposure duration was between 5 s and 10 s.

The linear extrapolation of the damage probability data yielded the 0% probability and 100% probability threshold values for the laser power and intensity ($P_{th0}$ and $P_{th100}$; $I_{th0}$ and $I_{th100}$), respectively (Table 3).

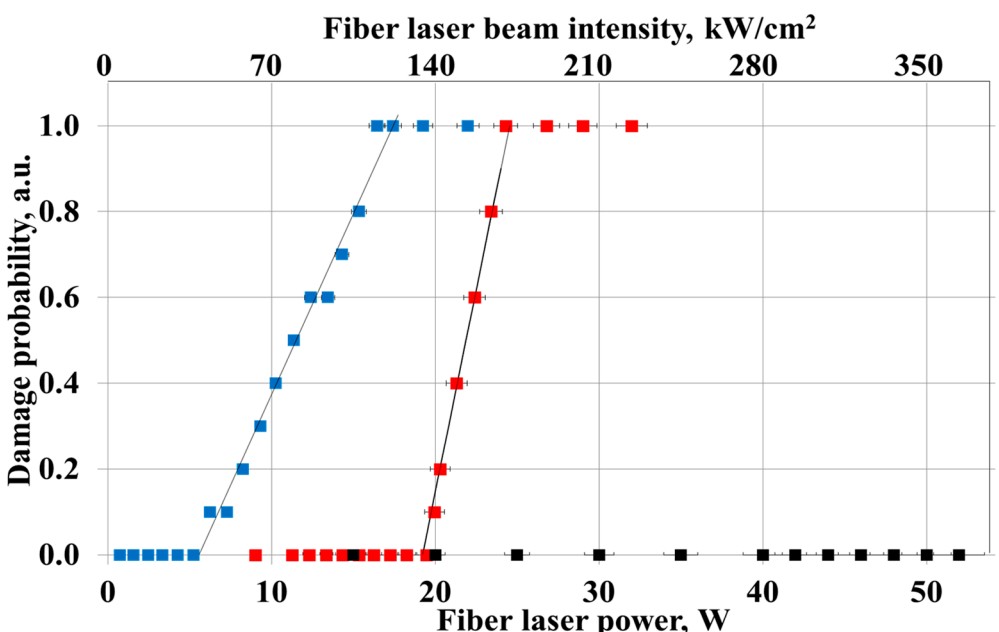

**Figure 6.** Damage probability vs. incident fiber laser power for $Ge_{20.5}As_{22}Se_{58}$ (red), $Ge_{35}As_{10}S_{55}$ (blue) and $As_2S_3$ (black).

**Table 3.** The LIDTs of ChGs.

|  | $Ge_{35}As_{10}S_{55}$ | $Ge_{20.5}As_{22}Se_{58}$ | $As_2S_3$ |
|---|---|---|---|
| Zero-probability LIDT for incident power/intensity, $P_{th0}/I_{th0}$, (W/W·cm$^{-2}$) | 5/35 | 19/133 | Nondetermined |
| 100%-probability LIDT for incident power/intensity, $P_{th100}/I_{th100}$ (W/W·cm$^{-2}$) | 17/117 | 24/168 | Nondetermined |

Photographs of the damaged chalcogenide samples are shown in Figure 7. They were captured with a HYPERION 2000 Bruker IR microscope (Germany) in the reflection and transmission modes using a 15× IR lens with an aperture of NA = 0.4.

Importantly, in all cases, damage was initiated on the output surface of the sample and the damage crater had a conical form (Figure 7). The LIDT difference on the exit and entrance surfaces could be explained by two effects: thermal self-focusing (the thermal lens formation) in the ChGs or a difference in the maximum electric field amplitude of the optical waves at the surfaces due to a phase difference of the waves reflected from the boundaries [43]. The second effect appears to be more reasonable for the thin transparent discs.

The damage spots, despite common damage thermal mechanisms described for sulfur- and selenium-containing glass [6–8], appeared dramatically different between samples. As can be seen in Figure 7a,b, the crater in the $Ge_{35}As_{10}S_{55}$ glass was surrounded by a dark area, possibly due to evaporation of highly volatile sulfur. A different type of formation with the characteristic metallic cluster can be seen around the crater induced in selenium-containing glass (Figure 7c). The chemical composition of substances around the craters will be studied in more detail in future.

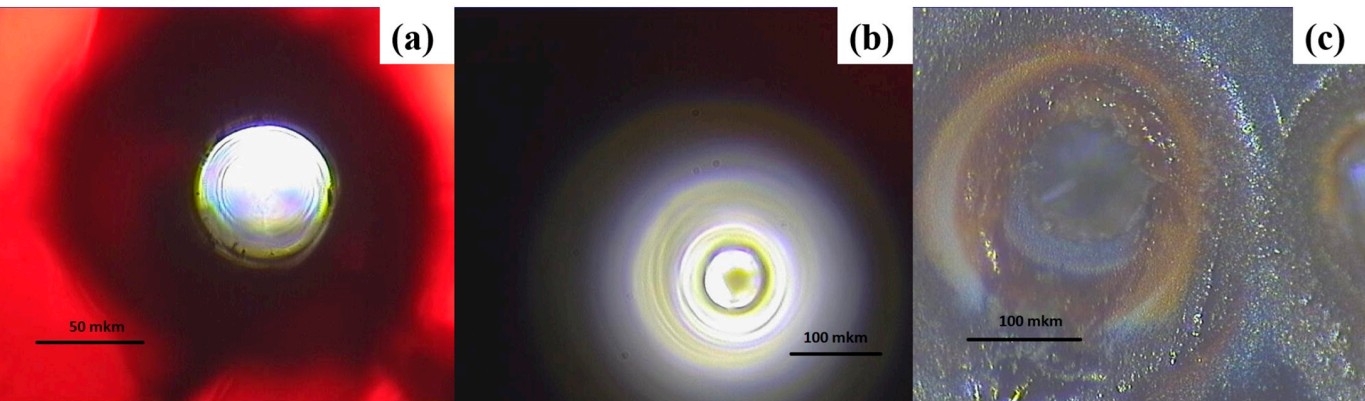

**Figure 7.** Photos of craters in glasses. $Ge_{35}As_{10}S_{55}$– views in transmitted light (**a**) and in reflected light (**b**); $Ge_{20}As_{22}Se_{58}$– view in reflected light (**c**).

## 4. Discussion

The nonlinear thermal refractive index $n_{2T}$ of $Ge_{35}As_{10}S_{55}$ and $Ge_{20}As_{22}Se_{58}$ glasses was found to be lower than that of $As_2S_3$ glass. However, thermal LIDT of the $Ge_{35}As_{10}S_{55}$ and $Ge_{20}As_{22}Se_{58}$ glasses appeared to be much lower than that of $As_2S_3$. This finding contradicts recently published papers in which an increase in the Ge concentration led to higher LIDT, at least in the case of the femtosecond repetitively pulsed irradiation [8–10,34,44].

To explain this discrepancy, the following should be noted. Both $Ge_{35}As_{10}S_{55}$ and $Ge_{20}As_{22}Se_{58}$ samples have a non-stoichiometric composition depleted in chalcogen (sulfur or selenium). However, the content of germanium is significantly higher in the $Ge_{35}As_{10}S_{55}$ sample. As follows from the publications mentioned above, the laser damage threshold is high for stoichiometric compositions [8,34,44], i.e., in our case, the glass compositions were not optimal for achieving the maximum possible LIDT. At the same time, despite the significant difference in $T_g$ (Figure 2), $Ge_{35}As_{10}S_{55}$ and $Ge_{20}As_{22}Se_{58}$ glasses exhibited approximately the same parameters of a thermal lens induced by the laser radiation at 1908 nm. We have obtained a higher LIDT in the $Ge_{20}As_{22}Se_{58}$ glass, compared to the $Ge_{35}As_{10}S_{55}$ glass. This behavior of LIDT can be related to the deviation from stoichiometry (R) [45]. At values of R > 1, the glass matrix was rich in chalcogens; at values R < 1, the matrix was depleted in chalcogens; at R = 1, the composition had an equilibrium state between the chalcogen atoms and other components. The values of the R parameter for $Ge_{35}As_{10}S_{55}$ and $Ge_{20}As_{22}Se_{58}$ glasses were 0.647 and 0.7945, respectively; these glasses were depleted in chalcogen, and the $Ge_{35}As_{10}S_{55}$ glass had a stronger deviation from stoichiometry. This agrees with the data [8,39], where it was experimentally established that the laser damage threshold decreases with a deviation from the stoichiometric glass compositions.

The optical band gap energy of $Ge_{20}As_{22}Se_{58}$, $Ge_{35}As_{10}S_{55}$ and $As_2S_3$ glasses is known to be 1.7, 2.37, and 2.35 eV, respectively [32,46,47]. The laser wavelength 1908 nm corresponds to 0.65 eV quantum energy, which is much less than the half-band gap of the studied ChGs. In this case, the inter-band absorption of the laser radiation could be caused by three or four quanta; however, the probability of such multi-photon transitions is negligibly small. Therefore, in our experiments, the difference in the LIDT didn't correlate with the band gap difference of the materials. On the other hand, it is obvious that the main absorption mechanism of laser radiation at 1908 nm, in the studied ChGs, is absorption from the energy levels of impurities and defects. The concentration of impurities in our special pure glass was very low. Defects in the glass structure were associated, in particular, with the presence of "dangling bonds" of the molecular network. Based on the known concepts of photo-induced chemical and structural changes in ChGs, it can be concluded that regions of positive and negative charge (the so-called "charged defects" [48,49]) are formed near the dangling bonds. The number of such defects increased with the deviation from the stoichiometry of the chemical composition. Therefore, the LIDT was the highest in stoichiometric $As_2S_3$ glass with low defect concentration, and the LIDT decreased

as the deviation from stoichiometry increased (and defect concentration increase) from $Ge_{20}As_{22}Se_{58}$ to $Ge_{35}As_{10}S_{55}$.

Another possible reason for the low LIDT of $Ge_{35}As_{10}S_{55}$ glass is that its germanium-enriched composition was located near the boundary of the glass formation area and was prone to crystallization [18,50], in contrast to $Ge_{20}As_{22}Se_{58}$ glass, which did not show crystallization peaks in DSC analysis (Figure 2). Resulting from the impact of a laser beam, local crystallization of $Ge_{35}As_{10}S_{55}$ glass is quite possible and it could lead to lower LIDT values.

The results of our measurements show that both the fundamental characteristics and specific properties of glass samples have an effect on the damage threshold. The nature of various factors, determined by the glass properties, and their influence on the laser damage resistance requires additional studies.

**5. Conclusions**

The thermal lens was measured in three ChGs using the Z-scan technique, with the quasi-CW fiber laser at 1908 nm. The on-axis phase shift and the steady-state nonlinear thermal refractive index $n_{2T}$ were determined for the $Ge_{35}As_{10}S_{55}$, $Ge_{20}As_{22}Se_{58}$, and $As_2S_3$ glasses. The laser-induced damage in ChGs under the fiber laser irradiation was tested following the one-on-one procedure. The thermo-optical nonlinear parameters were comparable between the tested ChGs samples; however, the LIDT value of $Ge_{20}As_{22}Se_{58}$ glass was found to be higher compared to that of $Ge_{35}As_{10}S_{55}$ glass. Both $Ge_{35}As_{10}S_{55}$ and $Ge_{20}As_{22}Se_{58}$ glasses had much lower LIDT than $As_2S_3$ glass. This difference in LIDT can be attributed to deviation from stoichiometry of glass composition and tendency to crystallization of the Ge-rich non-stoichiometric compositions.

Further research is needed to optimize the composition of Ge-As-S and Ge-As-Se glasses to provide the highest possible thermal LIDT and assist the development of optical media capable of transmitting high-power infrared radiation.

**Author Contributions:** Conceptualization, O.A., V.S. and E.K.; methodology, O.A., V.S. and E.K.; software, Y.G. and A.D.; validation, T.K. and E.K.; formal analysis, Y.G. and A.D.; investigation, A.D., O.A., Y.G., E.K. and M.S.; resources, M.S., T.K., E.K. and O.A.; data curation, E.K.; writing—original draft preparation, O.A., Y.G., E.K. and V.S.; writing—review and editing, O.A. and V.S.; visualization, Y.G.; supervision, O.A. and V.S.; project administration, O.A.; funding acquisition, O.A. All authors have read and agreed to the published version of the manuscript.

**Funding:** This research was supported by the Russian Science Foundation (project no. 22-12-20035, https://rscf.ru/project/22-12-20035/ (accessed on 27 February 2023)).

**Institutional Review Board Statement:** Not applicable.

**Informed Consent Statement:** Not applicable.

**Data Availability Statement:** Not applicable.

**Conflicts of Interest:** The authors declare no conflict of interest. The funders had no role in the design of the study; in the collection, analyses, or interpretation of data; in the writing of the manuscript, or in the decision to publish the results.

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
