# Peer review of "Thermal Lensing and Laser-Induced Damage in Special Pure Chalcogenide Ge35As10S55 and Ge20As22Se58 Glasses under Quasi-CW Fiber Laser Irradiation at 1908 nm"

_photonics, doi:10.3390/photonics10030252_

Round 1

Reviewer 1 Report

See attached

Author Response

REVIEWER 1.

COMMENT 1. It would be useful to refer in the Introduction section to the fundamental study [M Sparks, J. Appl. Phys. 42, 5029 (1971)] of optical distortions in optical windows, including chalcogenides, heated by laser radiation.

Answer: The information about the optical distortions by heated chalcogenide windows and the reference to the Sparks fundamental paper were added in the Introduction of our paper.

COMMENT 2. For better perceiving of the work, it would be useful to move the information given in lines 255-259 and Table 2 to around line 195, where that information required.

Answer: The information given in lines 255-259 (the lines 258-262 in the revised version of the paper) is the result of determination of the n2T parameter by our experiments. For this reason, the information was done at the end of the experiment description. No reason to move this information to another lines.

COMMENT 3. On line 217 the parameter is the fraction of the absorbed energy converted into heat but no information given what does it mean physically and what its magnitude in the current work.

Answer: The ξ -parameter name was changed from “the fraction of the absorbed energy converted into heat” to “the fraction of the absorbed power converted into heat”.  The above definition of the parameter appears to be physically self-sufficient, not requiring additional explanations. The parameter value was used as a multiplier of the absorption coefficient for the theoretical estimation of the n2T parameter, for example, for As2S3 glass (line 267): αξ = 1.5×10-3 cm-1.

COMMENT 4. The fact that the damage threshold at the exit surface of optically transparent materials is notably lower than that at entrance one is known for more than half a century and it is accounted for by M Crisp, IEEE J. Quantum Electron. QE-10, 57 (1974). The proposed on lines 301-302 explanation of the observed results does not look reasonable because of small thickness of samples.

Answer: We agree that the LIDT difference at the exit and entrance surfaces of optically transparent discs can be caused by the difference of the maximum electric field amplitude of the optical waves at the surfaces due to a phase difference of the wave reflected from the boundaries. This effect for thin transparent disks could be more severe than a thermal lens. We added this additional explanation of the experimentally-measured LIDT difference and the reference to Crisp paper.

Lines 313-318: “The LIDT difference on the exit and entrance surfaces could be explained by two effects: thermal self-focusing (the thermal lens formation) in the ChGs or a difference of the maximum electric field amplitude of the optical waves at the surfaces due to a phase difference of the waves reflected from the boundaries [44]. The second effect appears to be more reasonable for the thin transparent discs.”

Reviewer 2 Report

1. It appears in Fig. 5 that the parabolic thermal lens model fits the measured data clearly better than the aberrant lens model. However, in lines 238-242, it is stated "the aberrant lens model was found to give the better prediction of the parameters for strong thermal lensing." These two seem to contradict each other.   

2. The two glasses under inspection differ in optical bandgap energy. Even though the excitation wavelength seems to correspond to energy less than half the optical bandgap energy of the glasses, it is noteworthy to be mentioned in the text.

3. Some typos; e.g., Raileigh in line 214, Zerro in Table 3, Ge20.5 in Table 3, CHGs in line 135 

Author Response

COMMENT 1. It appears in Fig. 5 that the parabolic thermal lens model fits the measured data clearly better than the aberrant lens model. However, in lines 238-242, it is stated "the aberrant lens model was found to give the better prediction of the parameters for strong thermal lensing." These two seem to contradict each other.

Answer: We agree that the used thermal lens models (both parabolic and aberrant) didn’t give the fine predictions of the experiments. For this reason, we removed the following sentence part: ”the aberrant lens model was found to give the better prediction of the parameters for strong thermal lensing”.  

COMMENT 2. The two glasses under inspection differ in optical bandgap energy. Even though the excitation wavelength seems to correspond to energy less than half the optical bandgap energy of the glasses, it is noteworthy to be mentioned in the text.

Answer: We agree with the comment about the bandgap difference. However, the difference isn’t matter in the studied ChGs under 1908 nm pumping. To explain the absorption effect to LIDT the following paragraph and references were added to the text:

Lines 353-369: The optical band gap energy of Ge20As22Se58, Ge35As10S55 and As2S3 glasses is known to be 1.7, 2.37, and 2.35 eV, respectively [47-49]. The laser wavelength, 1908 nm, corresponds to 0.65 eV quantum energy, that is much less than half band gap of the studied ChGs. In this case, the inter-band absorption of the laser radiation could be caused by 3 or 4 quanta, however the probability of such multi-photon transitions is negligible small. Therefore, in our experiments, the difference in the LIDT don’t correlate with the band gap difference of the materials. On the other hand, it is obvious that the main absorption mechanism of laser radiation at 1908 nm in the studied ChGs is absorption from the energy levels of impurities and defects. The concentration of impurities in our special pure glass is very low. Defects in the glass structure are associated, in particular, with the presence of "dangling bonds" of the molecular network. Based on the known concepts of photo-induced chemical and structural changes in ChGs, it can be concluded that regions of positive and negative charge (the so-called “charged defects” [50,51]) are formed near the dangling bonds. The number of such defects increases with the deviation from the stoichiometry of the chemical composition. Therefore, the LIDT is the highest in stoichiometric As2S3 glass with low defect concentration, and the LIDT decreases as the deviation from stoichiometry increases (and defect concentration increase) from Ge20As22Se58 to Ge35As10S55.

COMMENT 3. Some typos; e.g., Raileigh in line 214, Zerro in Table 3, Ge20.5 in Table 3, CHGs in line 135.

Answer: The typos were corrected.

Author Response

COMMENT 1. After reading the article, there is a very contradictory impression about her. The article contains a lot of data on various properties of the studied glass samples, which, in my opinion, very poorly relate to the main topic given in the title of the article, which because of this turned out to be highly diluted in terms of the novelty of the data obtained. This part can be significantly reduced without any losses.

Answer: The present article describes the study of Ge35As10S55 and Ge20As22Se58 glasses as cladding materials for the preparation of active fibers with a core glass doped with Bi or rare earth ions (REIs), i.e. as laser materials. Therefore, the main purpose of the work was to determine the thermal nonlinear refractive index and laser induced damage threshold. Other investigations (optical transmission in the mid-IR region, content of optically active impurities, glass transition temperature and crystallizing ability) were needed to characterize these glass samples.

COMMENT 2. In the article, the objects of research are called glasses, and not samples of glasses that are being investigated. It is necessary to have great courage to generalize the measurements of the parameters of one available sample to all samples of the same composition... The article would be much more professional if it examined several samples of different compositions of both systems.

Answer: Indeed, in this article, the objects of research are glass samples of a given composition (Ge35As10S55 and Ge20As22Se58 glasses) with a given impurity content. In glass samples with the same composition, but with different concentrations of impurities, the studied properties (thermal nonlinear refractive index and laser induced damage threshold) may differ, but only slightly. We agree that the article would have been better if it had studied a series of glasses of different compositions from each system. But this was not included in the goals of this article, but will be the subject of research in our next papers.

COMMENT 3. In the description of the methods for measuring nonlinear properties, it is not highlighted what the authors of the article have introduced into them. If there is nothing new in them, this part can also be shortened, leaving the most important  links.

Answer: The descriptions of the experimental methods of the thermal lens testing and the laser induced damage threshold determination appears to be the important parts of our paper. The additional shortening of the text in these parts appears to be unreasonable.

COMMENT 4. There is no explanation in the article why such glass compositions (novel Ge35As10S55 and Ge20As22Se58 glass) were chosen and why the sulfide composition is new, although the diagrams of glass regions in triple Ge-As-S(Se) systems have been well studied for a long time.

Answer: We used the Ge35As10S55 and Ge20As22Se58 glass compositions as a cladding material for fabrication of step-index REIs(or Bi)-doped-core optical fibers due to their suitable optical and thermal properties, glass transition temperature, refractive index, as well as low adhesion to silica glassware (that is important in obtaining high-quality glass rods). We have chosen these compositions experimentally from a series of glasses (which is not the subject of this article) to match the properties of the core glasses (Ga-Ge-As-Se and Ga-Ge-Sb-Se systems), to ensure the required numerical aperture, viscosity, resistance to pump radiation and the achievement (in the future) of a high power of laser generation of radiation in the mid-IR range. At present, the authors of this work from the Institute of Chemistry of High-Purity Substances of the Russian Academy of Sciences have obtained Tb(3+)-doped fibers with laser generation at 5 μm with a power of 150 mW [V.V. Koltashev, B.I. Denker, B.I. Galagan, , V.G. Plotnichenko, G.E. Snopatin, M.V. Sukhanov, S.E. Sverchkov, A.P. Velmuzhov, 150 mW Tb3+ doped chalcogenide fiber laser emitting at ~5 microns, Opt. Laser Technology. (in press)]. To optimize optical fibers and achieve higher lasing power, cladding glasses with a high laser breakdown threshold are required.

COMMENT 5. On page 3 (Fig. 1), the authors give the "absorption spectra" of the studied glasses obtained from their transmission spectra T. At the same time, they do not describe the procedure for calculating absorption. The spectra shown in Fig. 1 are at best -ln T/d (or -logT/d), where d is the thickness of the sample, i.e., up to a constant, this is the attenuation coefficient. This also does not take into account the change in the refractive index of the glass, which can vary quite a lot in such a wide range of measurements.

Answer: The absorption spectra were calculated based on the fact that when radiation propagates in the sample, its intensity decreases exponentially in accordance with the Bouguer-Lambert-Behr law I=Io∙exp(-α∙d), where Io is the intensity of the incident wave; I is the intensity of the transmitted wave, d is the sample thickness; α absorption coefficient.

            Then, if we neglect the reflection from the sample surface, to obtain the absorption spectrum, we can use the ratio -lnT/d, where T=I/Io is the transmission.

For a more rigorous approach, it was necessary to measure not only the transmission spectrum, but also the reflection spectrum, which we could not measure due to the lack of a special attachment. The dispersion of the refractive index was not taken into account in the calculations, since it is not exactly known for the glass compositions used in the work. An analysis of the literature data for the refractive index of glasses of close macrocomposition showed that changes in the content of one of the macrocomponents, even by one atomic %, lead to a significant change in the refractive index, in the first and second decimal values.

            In this work, IR spectra were recorded to control the impurity composition of glasses. The impurity content was measured from the peak intensity of the absorption bands relative to the zero baseline; therefore, the method used to calculate the absorption spectra did not affect the accuracy of their determination.

COMMENT 6. "The method of IR spectrometry does not allow to determine the background level of the absorption coefficient of glasses" is true only for modern routine devices in which the measured sample is illuminated in the focus of the sample camera and radiation receivers with a small sensitive area are used. For accurate measurements of attenuation coefficients, parallel beam attachments and receivers with a large sensitive area or integrating spheres are used.

Answer: In our work, the registration of IR spectra was carried out in the cuvette cell of the Tenzor-27 (Bruker) IR spectrometer, for routine measurements, using a DLATGS receiver with a 2x2mm area, not equipped with additional attachments for measuring the absolute value of reflection, measuring the absolute value of transmission and integrating sphere. 

COMMENT 7. As for the spectra themselves, the presence of absorption bands in them associated with fluctuations in carbon, water and C-H bonds with an intensity >=0.01 cm-1 (4.3 dB/m) does not allow these glasses to be classified as high-purity. The presence of intense S-S bond oscillation bands indicates a deviation of the composition of the «As2S3» sample from the stoichiometric one.

Answer: We agree with the remark on the terminology of glass purity. It is more correct to call the studied glasses as special pure. The spectrum of As2S3 glass (Fig. 1a) was re-measured, and the CO2 band, which appeared due to the measurement in air, was eliminated (compensated). The chemical composition of the sample was confirmed by inductively coupled plasma atomic emission spectrometry. In the spectrum of this glass, the bands at 6.8 µm and 7.56 µm do not belong to the S-S bonds, but to the overtones of the As-S bonds, according to [M.S. Maklad, R.K. Mohr, R.E. Howard, P.B. Macedo and C.T. Moynihan Multiphonon absorption in As2S3-As2Se3 glasses, Solid State Communications. - 1974. - V. 15. - P. 855-858.] and [S. Tsuchihashi, Y. Kawamoto, Properties and structure of glasses in the system As-S, Journal of Non-Crystalline Solids V.5, Issue 4, 1971, 286-305]

A few additional remarks:

line 35-37:       «Recently, lasing in the 4.5-5.9 pm wavelength range was demonstrated in glass samples and optical fibers based on ChGs doped with Ce3+, Tb3+, Dy3+ ions [4,5]».

  • I could not find in the articles [4,5] reports on generation in activated Dy3+ chalcogenide

Answer: We agree with the reviewer and have replaced Dy3+ with Pr3+.

line 77:     «The Ge35As10S55 and Ge20As22Se58 types of glass with a high content of germanium were chosen for the laser damage resistivity studies due to their sufficiently high glass transition temperature and improved thermal characteristic.»

  • What do the authors mean by the types of glasses and what are the thermal characteristics of these glasses better than others?

Answer: The phrase "The Ge35As10S55 and Ge20As22Se58 types of glass with a high content of germanium were chosen for the laser damage resistivity studies due to their sufficiently high glass transition temperature and improved thermal characteristic." was corrected to: “The Ge35As10S55 and Ge20As22Se58 glasses with a high content of germanium were chosen for the laser damage resistivity studies due to their sufficiency high values of glass transition temperature and high stability against crystallization”.

line 142:    «However, the presence of crystallization peaks cannot be ruled out at temperatures higher than what is allowed by the DSC experimental conditions».

  • It is not clear what the authors mean by acceptable conditions and why this phrase is given in the

Answer: This phrase was excluded.
